# Comparison of INTEGRA and the Manual Method to Determine the Axis for Intraocular Lens Implantation—A Case Series of 60 Eyes

**DOI:** 10.3390/healthcare10091773

**Published:** 2022-09-14

**Authors:** Marcin Jaworski, Dorota Wyględowska-Promieńska, Piotr Jaworski, Michał Kowalski, Krzysztof Jaskot, Robert Bieda

**Affiliations:** 1Ophthalmological Center for Children and Adults Optomed, 41-500 Chorzów, Poland; 2Department of Ophthalmology, School of Medicine in Katowice, Medical University of Silesia, 40-055 Katowice, Poland; 3Department of Ophthalmology, Prof. K. Gibiński University Clinical Center of Medical, University of Silesia in Katowice, 40-007 Katowice, Poland; 4SalMed Michał Kowalski, 91-502 Łódź, Poland; 5Department of Automatic Control and Robotics, Silesian University of Technology, 44-100 Gliwice, Poland

**Keywords:** computer-assisted surgery, digital assisted surgery, axis designation, integra, intraoperative device, axis determination, automated surgical support system, toric lens

## Abstract

(1) Background: To compare the results of a new intraoperative contactless device (INTEGRA Optomed, Poland) with the result of a manual method for determining the axis for toric intraocular lens implantation. (2) Material and Methods: This retrospective observational study included 60 eyes of 40 patients (17 men, 23 women) who had toric intraocular lenses implanted. A video recording of each surgery that used the INTEGRA system was made for the analysis. Two researchers then independently assessed the location of the implant axes determined with both digital and manual slit-lamp methods, and compared the results between methods. (3) Results: The implantation axes suggested through the manual and INTEGRA methods were similar. The median axis disparities were 0.0 degrees for both groups. The standard deviation was 0.63 and 0.75 for researcher 1 and 2, respectively. The dominant value was 0.0 in both groups. The INTEGRA axis designation was statistically significantly different from the manual method for researcher 1 (*p* < 0.05), but it was statistically insignificant for researcher 2 (*p* = 0.79). (4) Conclusions: The INTEGRA system is a digital ink-free device for image tracking scleral vessels. It was helpful for determining the implantation axis in a precise manner, and the measurements were comparable with those obtained through a manual technique.

## 1. Introduction

Major changes have been occurring in the field of preoperative preparation and the performance of procedures for the use of toric intraocular lenses for the past several years. The changes are aimed at obtaining the best possible postoperative visual acuity, which in turn leads to an increasing use of toric lenses, even at low astigmatism values [1,2]. The introduction of such lenses has led to the creation of a diverse set of instruments to support surgeons in determining the axis of implanted lenses. There are numerous manual methods for determining the axis of implanted intraocular lenses. Automatic systems for determining the axis of the implant are becoming increasingly common. Such methods involve overlaying an image on the eye in real time to guide the incision site and capsulorhexis and they use the Technischer Ausschuss für BrillenOptik (TABO) scale to facilitate axial positioning of the intraocular lens in the eyeball [3,4] (Figure 1).

Proper toric lens alignment has a significant effect on postoperative uncorrected visual acuity. A study found that, for each degree of misalignment, astigmatism correction was 3.3% worse [5]. Additionally, the cyclotorsia of the eyeball influences the position of the intraocular lens in relation to the calculated axis of the implant [6]. Currently, there are three image-guided axis designation systems: the Alcon Verion Image-Guided System (Alcon Laboratories, Inc., Fort Worth, TX, USA), the Zeiss Callisto Eye and Z align (Carl Zeiss Meditec AG, Dublin, CA, USA), and TrueVision 3-D (3-dimensional) Surgical System (TrueVision Systems, Inc., Santa Barbara, CA, USA) [3,7,8,9,10]. These systems have been introduced recently and research is currently being performed to verify their usefulness and effectiveness in determining the axis of implantation relative to established methods.

The INTEGRA system uses a similar foundation for determining axis designation as the previously mentioned image-guided systems. The INTEGRA system reported here is based on defining the eyeball mathematically, and thus determining the implant axis according to characteristic points indicated by the digital image in real time.

The region of interest is an active area, where the system tracks an image. Obscuring the region of interest by instruments or impermeable fluids (e.g., blood), deformation or rotation of the eyeball can decrease the effectiveness and stability of the tracking. The lid speculum deformations were ignored. The following key points in tracking were proposed: scleral vessels and a ring segment, restricted internally by the cornea and externally by the eyelids (Figure 2).

Vessels narrowed by pharmacotherapy could still be used, however, as long as their outline remains visible. In accordance with performed preliminary tests, the tracking remains stable enough. The scleral vessels, unlike conjunctival ones, do not constrict and their outlines remains visible. The scleral vessels provide a stable and repeatable designation that is not affected by blurred ink or blood obscured marks [7].

This article is a continuation of the work on the intraoperative image-guided system for determining the axis of the INTEGRA system [3,7]. This study was designed to explore the effectiveness and accuracy of a new device, INTEGRA, the characteristics of which are described below, and compare results with the measurements made manually. It is based on a multi-center study assessing the accuracy when determining the axis of the implant in relation to the manual method using a slit lamp and a marker. The system is connected to the visual path of the operating microscope and tracks the scleral vessels in real time. Based on this tracking, the system orientates the TABO scale and displays information to the surgeon.

## 2. Materials and Methods

This study was a multicenter, retrospective, observational study. The INTEGRA system is noninvasive, serves as a tool to check the position of the implant axis and uses the TABO scale in relation to reference methods.

The INTEGRA system is based on an algorithm introduced by D. Lowe, the Scale Invariant Feature Transform [11]. The system tracks scleral vessels in the region of interest and detailed image analyses were previously described [3,7]. Notably, the system allows tracking of the vessels and superimposes the image regardless of the eyeball position, including cyclotorsia, in real time. The graphs representing the TABO scale, the flat and steep corneal meridians and the implant axis, as well as the location of the main incision, side-port incisions and the visual axis, are superimposed on the image from the operating microscope [3,7]. The INTEGRA is an open structure system. To work, it requires an image from any operating microscope and a reference image—a photo of the anterior segment of the eye and corneal topography. The central unit (computer) along with dedicated software processes the images. Reference images as well as planned intraocular lens axis and other parameters, e.g., visual axis, are introduced from the corneal topography. A schematic diagram of the operation of the system is shown in Figure 3.

The INTEGRA system eliminates misalignments arising from the parallax error due to shifting of patient’s head in relation to the operating microscope, cyclotorosion of the eyeball or any obscuration of the reference (operated) area. Moreover, it enables working in the dim light of an operating field [3,7]. Two researchers (the surgeon was excluded from this stage) independently assessed the location of the designated implant axes using both methods and then compared the results between methods. To maintain impartiality, researcher 2 was affiliated with a different institution than the surgeon. The device described here has been used in phacoemulsification procedures with implantation of toric intraocular lenses as well as toric phakic lenses.

Following completion of corneal topography with an Oculus Keratograph III (Oculus, Wetzlar, Germany) and anterior segment photography, the system could recognize tens of thousands of reference points on the sclera and determine the visual axis. The TABO scale is centered on the pupillary center comparing to reference image of scleral vessels in the region of interest. Visual axis coordinates were exported as the center of Placido rings from the topography. The visual axis is presented by the system to the surgeon. As the final step, the TABO scale was applied to the image (Figure 4) [3].

Images that were analyzed in the current study were from surgeries in 2017 and 2018 at the Ophthalmological Center for Children and Adults, Optomed, Chorzów, Poland. Before the surgery, a surgeon experienced in toric intraocular lens implantation (PJ) manually marked in a single step the implant axis with a surgical pen, using a slit lamp. An OMS 800 operating microscope (Topcon, Tokyo, Japan) was used, and video recording of the surgery was analyzed using the INTEGRA system. Images from surgery are presented in Figure 5. The exclusion criteria for further INTEGRA analyses were blurry image, iris and conjunctiva not fully visible in the footage, conjunctival hemorrhage covering more than half of the eye, non-toric intraocular implant and lack of patient consent for further analyses.

### 2.1. Sample Size Calculation

Sample size was estimated using G*Power (version 3.1.9.4., Heinrich-Heine-Universität, Düsseldorf, Germany). To obtain power 0.85 with alpha 0.05 at effect size 0.315, minimal sample size was 55.

### 2.2. Statistical Analysis

The statistical analysis was conducted using Statistica Software version 7 (StatSoft, Inc., Tulsa, OK, USA) and Microsoft Excel version 16.20 (Microsoft Co., Redmond, WA, USA). The Shapiro-Wilk normality test was applied. Wilcoxon matched pairs test was used to compare paired measurements for two researchers, Friedman test was performed to compare more than 2 groups of dependent measurements. Bland-Altman method was used to compare analysed methods and to analyze associations between quantitative variables Pearson’s correlation coefficient was performed.

### 2.3. Ethical Consideration

Pursuant to the decision of the Bioethics Committee of the Medical University of Silesia in Katowice, Poland No. KNW/0022/KB/36/19, the study was observational and did not involve medical experiments. Therefore, the method did not require any evaluation by the Bioethical Committee. This study adheres to the tenets of the Declaration of Helsinki. The patients provided signed consent for the use of images for further analysis by the INTEGRA system.

## 3. Results

Sixty eyes of 40 White patients (17 men, 23 women) who had toric intraocular lens implantation (phakic implants or combined with cataract phacoemulsification) were included in the study. The age range of the patients was 22–78 years, with an average age of 46 years. The group that had phakic implantation included 12 women (20 eyes) and 8 men (13 eyes), aged 22–34 years (average 27 years). The group of 20 patients (27 eyes) having undergone a phacoemulsification procedure with primary toric intraocular lens implantation consisted of 11 women (15 eyes) and 9 men (12 eyes), aged 44–78 years (average 68 years) (Table 1).

The comparison was made to evaluate axis deviation between manual and digital marking. The median disparities of the designated intraocular lens axis were 0.0 for both groups and the standard deviations were 0.63 and 0.75 for researchers 1 and 2, respectively, in the comparison of the INTEGRA TABO scale view and the manually determined method. The dominant value of the designated intraocular lens axis to INTEGRA was 0.0 for both researchers. For phakic surgeries, the standard deviation (axis designated manually vs. INTEGRA) was 0.52 for researcher 1 and 0.68 for researcher 2 and, for pseudo-phakic surgeries, the standard deviation (axis designated manually vs. INTEGRA) was 0.84 for researcher 1 and 0.89 for researcher 2. Mean absolute deviation from planned implantation axis was 0.10 for surgeon, 0.32 for researcher 1 and 0.25 for researcher 2. The range of axis deviation was 2.0 and 3.0 for researcher 1 and 2, respectively. Measured data between groups (axis designated manually vs. researcher 1 and axis designated manually vs. researcher 2) had the same non-normal distribution (level of significance α < 0.05). Results of axis designated manually vs. researcher 1 was statistically significant (*p* < 0.05) and that of axis designated manually vs. researcher 2 was statistically insignificant (*p* = 0.79). The Pearson’s correlation coefficient between the manually determined axes was strong for both researchers (r > 0.99, *p* < 0.01 in both cases). Both researchers showed high correlation between their measurements and, based on the Bland–Altman plot, the mean difference was 0.2. Researcher 2 had greater deviation from the manual readings than researcher 1, especially for the oblique axes of implantation.

## 4. Discussion

This study showed that the INTEGRA system and manual determination yielded statistically comparable results for the implant axis. Both researchers participating in the study determined the axis of the implant similarly. There were discrepancies in the determined axis of the phakic and pseudo-phakic implant between researchers, which may have been due to a lack of depth in the displayed image as compared with the perception of depth during the procedure when using an operating microscope. Additionally, the researchers evaluating the implant axis did not know the calculated axis of the implant. The deviation of the implant axis reported by the researchers was close to the designated axis of the implant and the standard deviation of the determined axis did not exceed 1 angular degree. Researcher 2 designated the axes with a greater standard deviation, which may be explained by that researcher being from another medical clinic and not being as familiar with the equipment and operating techniques of the surgeon performing the procedure. That greater deviations led to the statistical insignificance of researcher 2′s results. Nevertheless, the axis deviation was still clinically acceptable. Additional statistical tests used in the analysis indicate a high correlation of the results. For this reason, the results can be considered reliable. From a practical point of view, the obtained measurements allow for precise alignment of the intraocular lens, which in turn provides greater chance of achieving therapeutic success, good uncorrected visual acuity and spectacle independence. The methods of manual determination of the implant axis are considered to be effective, precise and repeatable, therefore can be chosen as reference procedure [3,6,7,12].

Unfortunately, the inks used for determining the axis of the implant may fade or be erased, which can lead to significant errors in the postoperative axes of the implant [10]. Instead of ink, thermal marking of the axis on the cornea can be used (Thermodot, BVI, Waltham, MA, USA) [3]. Moreover, femtosecond laser-assisted lens capsular marking may also be used, which allows for more precise determination of the implant axis compared with the manual method. The laser was applied only for determining the axis, but not for breaking the lens nucleus or for capsulorhexis. Nevertheless, Chen and Zhang used the 0° to 180° axis determination of the implant by applying a needle and marker [13]. A Mendez gauge was next used to determine the implant target axis. This multi-stage process can lead to errors, which was noted by the researchers themselves. They determined the axis by using a narrow slit in the slit lamp and marking the axis of the implant with a marker. Laser incision can potentially lead to weakness and tears in the lens capsule, which may in turn result in perioperative complications or late instability of the implanted toric lens [13]. This approach is a combination of automatic and manual methods, but without real-time display of the marker image.

Another alternative to manual methods for determining the axis of the implant is to use markers attached to a Goldman tonometer (e.g., ToMark corneal marker, Geuder, Heidelberg, Germany), a gravity-leveled pendular handheld marker to place a reference point at the horizontal meridian (e.g., Whitehouse Gravity Axis Marker, Rumex, Tampa, FL, USA), or a bubble marker, which uses a bubble level to maintain the horizontal meridian in a handheld instrument (e.g., Nuijts-Lane Pre-op Toric Reference Marker With Bubble, ASICO, Westmont, IL, USA). The manually determined axis can rechecked and refined using an image displayed by a smartphone (toriCam application) [6,14]. Of the manual methods, the determination of the implant axis with the use of a slit lamp in a seated position is regarded as the most accurate [12]. We used this technique as the reference method. Despite this, there are increasing numbers of studies that confirm the superiority of automatic methods over manual ones [15,16].

There are solutions on the market that intraoperatively apply the TABO scale in real time and specify the location of the ports as well as the point where capsulorhexis is performed. These solutions include the Alcon Verion Image Guided System (Alcon Laboratories, Inc., Fort Worth, TX, USA), the Zeiss Callisto Eye (Carl Zeiss AG, Dublin, CA, USA) and the TrueGuide^®^ Computer-Guided Surgery, which is a 3D visualization system (TrueVision Systems, Inc., Santa Barbara, CA, USA) [4,9]. The first two require biometers created by the respective manufacturer, while the last system allows the importing of biometric results from external systems [9]. To our best knowledge, there is only one study directly comparing the Callisto and Verion systems, which was carried out by Hura and Osher [10].

In that study, the authors found that the difference in the determined axis between the systems did not exceed 3° in 53% of the analyzed images (16 eyes). The systems were found to be of similar accuracy and neither solution predominated over the other [10]. However, the study was not conducted intraoperatively, the results were not compared with a control group (i.e., manual determination of the implant axis), and the repeatability of axis determination was not assessed.

There have been few scientific reports regarding the accuracy of the TrueVision System. In one of the available publications, the system was compared with the manual method and yielded statistically similar results. The mean absolute errors were 2.96° ± 2.54° for the TrueVision System and 2.88° ± 2.18° for the manual method [9]. It is generally acknowledged that an implant axis deviation over 10° from the assumed is unacceptable [17].

In the current study, as well as in our previous one, the disparities of the designated axis were lower [3]. On the basis of a meta-analysis that compares image-based methods with manual methods of determining the implant axis, automatic methods were shown to be more precise and to allow for lower residual astigmatism [18]. In addition, they were faster to apply and carried no risk of infection when determining the axis and ink fading.

Our current and previous study [3] indicate that axis determination by using INTEGRA is accurate enough for daily practice as an alternative technique. The dominant value was 0.0 thus, in most of the cases, the system showed the same axis as the manual reference. The results of our studies lead to the conclusion that the INTEGRA has a similar repeatability and reliability of measurement as other image-guided systems. However, a direct comparison of all automatic methods of determining the axis of the implant will confirm the statement.

The phenomenon of cyclotorsia of the eyeball during a procedure may affect the setting of toric intraocular lenses. Usually, cyclotorsia is 4°, but it can sometimes be above 10°, and the deviation may be clockwise or counterclockwise. Changing the patientʼs position from sitting to lying leads to a 2° cyclotorsia [8]. It has been shown that both automatic and manual methods allow for reliable determination of the implant axis [18,19]. Therefore, in the INTEGRA system, reference images are collected in a sitting position.

Despite constantly improving calculation and operational techniques, a significant remaining problem is the postoperative rotation of lenses. During an uncomplicated cataract surgery, lens rotation is 1–3° within 24 months of surgery. Kramer et al. [20] observed that in patients with residual astigmatism, 77.4% of cases were due to intraocular lens rotation of ≥5°. The risk of rotation increases if the lens was originally aligned with oblique axes. It should be emphasized that the inclusion criterion in that study was postoperative residual astigmatism, and the data came from an online toric intraocular lens back-calculator. Initial rotational rates (>5°) for intraocular lenses were observed to range from 9.5% to 23% [20]. This outcome highlights the importance of positioning the lens as precisely as possible at the time of implantation.

The INTEGRA system uses scleral vessels for real-time tracking. During the procedure, despite medications used, they do not disappear, nor do they change significantly, in contrast to other structures such as the iris, which is untraceable during mydriasis. The algorithm used in the INTEGRA system showed accuracy and tracking stability [3,7]. The authors believe that the INTEGRA system is an alternative system to those already available on the market. Additionally, the feature is an open design, so the surgeon is not obliged to use the equipment of one manufacturer. This should improve the availability of image-guided systems worldwide.

Systems for determining the axis of the implant can support surgeons in their daily practice by offering an additional tool, free of defects and potential risks associated with the use of manual methods.

## 5. Conclusions

INTEGRA is a digital real-time intraoperative method of axis designation. The tracking is based on the scleral vessel image and it is statistically as precise as the manual slit-lamp technique. However, the image-guided method is contactless with no risk of markers fading.

## 6. Limitations

The determining the axis and the subsequent surgical results based on the INTEGRA system enables an outcome similar to that achieved with manual methods without the need for manual marking and the problems associated with that approach. Nevertheless, the results must be confirmed with randomized controlled trials with larger group of patients, and the procedures should be carried out by a larger number of surgeons. In addition, assessments should not only monitor the intraoperative implantation axis, but also include postoperative measurements with the patient in a seated position and recumbent. The authors are planning to use the system during operation procedures and base the intraocular lens implant axis on this. The study was a pilot project concerning the INTEGRA system. For this reason and due to technical difficulties of the study, no direct comparison with other commercially available automatic systems for intraoperative determination of the implant axis was made. In the future, the authors plan to compare with other available automatic systems.

## Figures and Tables

**Figure 1 healthcare-10-01773-f001:**
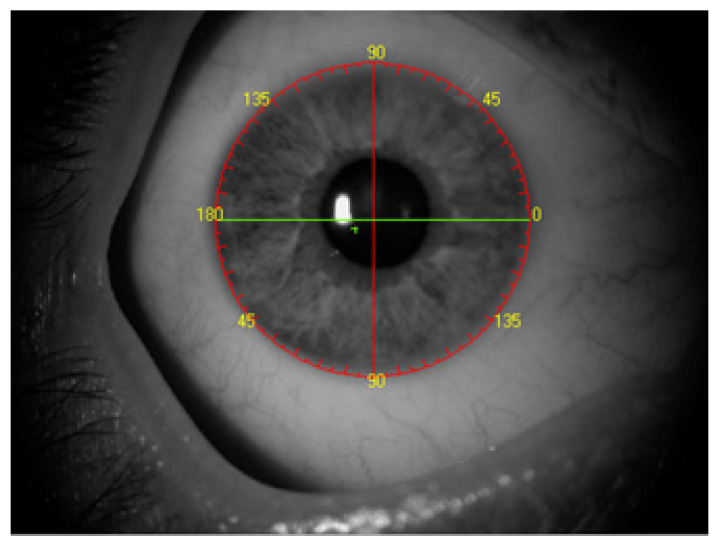
TABO scale.

**Figure 2 healthcare-10-01773-f002:**
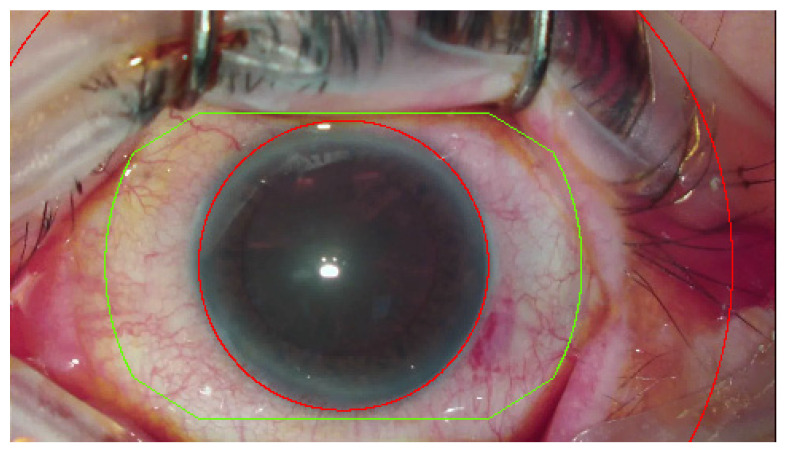
Region of interest of the INTEGRA system. Area indited to analyze is outside red circle and limited by green line.

**Figure 3 healthcare-10-01773-f003:**
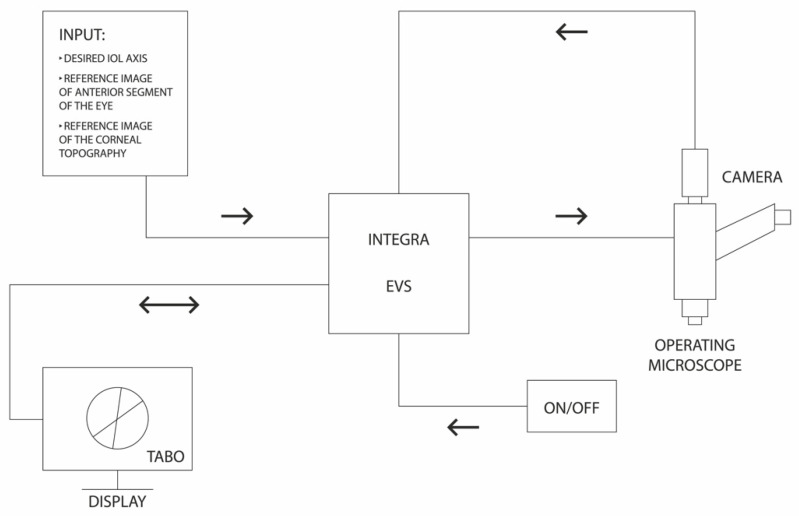
Schematic diagram of the operation of the INTEGRA system. EVS—computer platform on which the INTEGRA software runs.

**Figure 4 healthcare-10-01773-f004:**
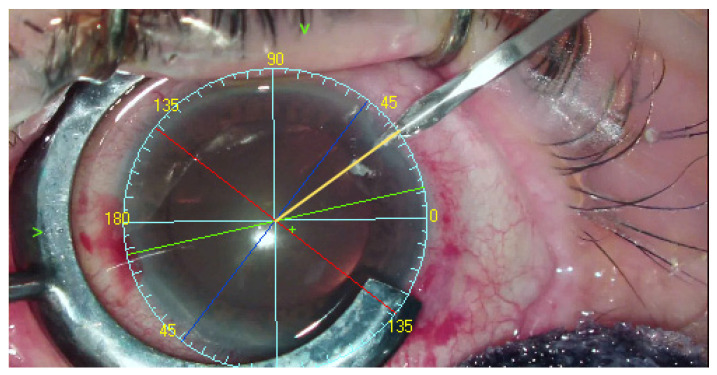
The INTEGRA system operation view: Blue lines and yellow numbers—TABO scale. Green cross—visual axis. Green arrows indicates the visual axis (not INTEGRA view), Dark blue line—flat meridian. Red line—steep meridian. Yellow line—main incision suggestion. Green line—intraocular lens implantation axis.

**Figure 5 healthcare-10-01773-f005:**
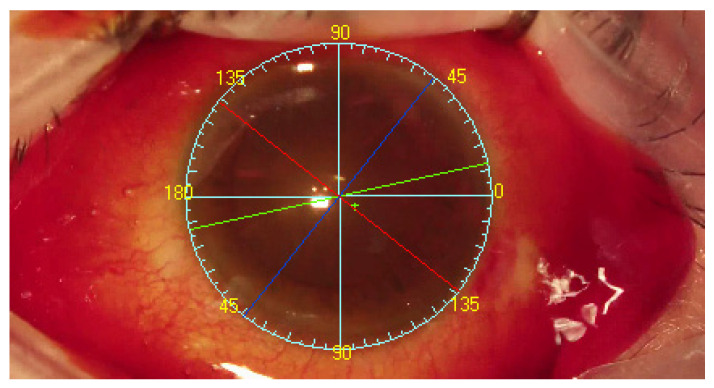
The INTEGRA system operation view. The system presents the TABO scale, despite the fact that a significant area of the region of interest is obscured by blood.

**Table 1 healthcare-10-01773-t001:** A demographic table of the study.

Parameters\Groups	Patients	Eyes	Range of Age [years]	Mean Age [years]
Female phakic	12	20	22–34	28.4
Female pseudo-phakic	11	15	44–77	67.3
Male phakic	8	13	22–30	25.4
Male pseudo-phakic	9	12	65–78	69.8
Total	40	60	22–78	45.8

## Data Availability

Original data available upon request.

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
