# Peer review of "Comparison of INTEGRA and the Manual Method to Determine the Axis for Intraocular Lens Implantation—A Case Series of 60 Eyes"

_healthcare, 2022, doi:10.3390/healthcare10091773_

Round 1
Reviewer 1 Report
The article presents the clinical experience with a digital corneal axis marking for toric IOL alignment and compares it to the slit-lamp assisted manual marking method.
1. Please elaborate on the system hardware and software
- what is the required equipment for a surgeon in order to use the system- slit-lamp camera or a different device, computer software etc.
- how is the desired IOL axis input is delivered to the system
- what device is connected to the ER microscope
2. Methods:
Paragraph 1, the exact number of patients included belong with the results section.
It is a bit confusing as there are 40 patients but details regarding gender of 60 eyes?
Both the manual and digital references were available to the surgeon during surgery, which one was preferred for IOL alignment in cases of disparity?
Paragraph 4, Please explain how the visual axis is determined by the system. Should the TABO scale be aligned with the corneal center/ pupillary center/ capsule center or the system’s visual axis for better refractive and visual results?
Please clarify if in the manual method, the desired IOL axis was marked on the slit lamp or the 180-0 horizontal reference followed by axis marking with a ring? i.e., one step or 2-step marking (which is more prone to misalignment).
3. Results:
Lines 161-165-
It is unclear what was compared with what.
Please explain what method was significant for 1 researcher and not for the second one.
Please clarify the comparison made: Manual vs. digital or manual marking vs. operative or postoperative IOL alignment and digital marking vs. IOL alignment?
4. Discussion
The added value of the Integra system compared with manual markings was fully explained. Is there an advantage to the system in comparison with other available digital marking systems on the market?
Line 211- The ToriCam does not replace eye-marking. The app identifies ink marking errors.
5. Typo errors:
Line 61- did you mean the ROI is an active area?
Line 62-63- unclear, please rephrase. Di you mean
Line 73- remain
Line 96- scleral
Line 146 were
Line 271 which is untraceable
Line 272, repeated “stability”
Reference 1 of (space) lens
Reference 6, 16 and 18 add additional authors
Author Response
Dear Reviewer,
Thank you for your time, detailed evaluation of the publication and comments. I have included responses to the comments below.
- Please elaborate on the system hardware and software
- what is the required equipment for a surgeon in order to use the system- slit-lamp camera or a different device, computer software etc.
- how is the desired IOL axis input is delivered to the system
- what device is connected to the ER microscope
We elaborate this topic. Additionally, we presented the figure to improve clarity of that paragraph.
- Methods:
Paragraph 1, the exact number of patients included belong with the results section.
Clarified
It is a bit confusing as there are 40 patients but details regarding gender of 60 eyes?
Clarified
Both the manual and digital references were available to the surgeon during surgery, which one was preferred for IOL alignment in cases of disparity?
Due to the study design, the surgeon operated using only the manual method. The INTEGRA system and the displayed image were available, but were ultimately analyzed after the procedure.
Paragraph 4, Please explain how the visual axis is determined by the system. Should the TABO scale be aligned with the corneal center/ pupillary center/ capsule center or the system’s visual axis for better refractive and visual results?
The TABO scale is centered on pupillary center comparing to reference image of scleral vessels in ROI (region of interest). Visual axis coordinates were exported as center of Placido rings from the topography. Visual axis is presented by the system to the surgeon. We aware that is an simplification and needs to be evaluated in other studies.
Please clarify if in the manual method, the desired IOL axis was marked on the slit lamp or the 180-0 horizontal reference followed by axis marking with a ring? i.e., one step or 2-step marking (which is more prone to misalignment).
Clarified. It was single step method.
- Results:
Lines 161-165-
It is unclear what was compared with what. - Clarified
Please explain what method was significant for 1 researcher and not for the second one. - Clarified
Please clarify the comparison made: Manual vs. digital or manual marking vs. operative or postoperative IOL alignment and digital marking vs. IOL alignment?
Clarified. It is manual vs digital marking.
- Discussion
The added value of the Integra system compared with manual markings was fully explained. Is there an advantage to the system in comparison with other available digital marking systems on the market?
At this stage, we believe that INTEGRA is an alternative system to those already available on the market. Additionally, the feature is the open structure of the system, the surgeon and the center are not forced to use equipment from one manufacturer. This should improve the accessibility of the image-guided systems.
Line 211- The ToriCam does not replace eye-marking. The app identifies ink marking errors.
The sentence was rephrased to omit generalization.
- Typo errors:
Line 61- did you mean the ROI is an active area? – yes, corrected.
Line 62-63- unclear, please rephrase. Di you mean – Rephrased.
Line 73- remain - corrected.
Line 96- scleral - corrected.
Line 146 were - corrected.
Line 271 which is untraceable - corrected.
Line 272, repeated “stability” - corrected.
Reference 1 of (space) lens - corrected.
Reference 6, 16 and 18 add additional authors - corrected.
Yours faithfully,
The authors

Reviewer 2 Report
Please clarify the advantages of using INTEGRA system compared to the other commercial devices.
‘Please consider adding a demographic table.
Please consider adding a paragraph to describe how this intraoperative device worked.
Please improve the resolution of all figures.
Author Response
Dear Reviewer,
Thank you for your time, detailed evaluation of the publication and comments. I have included responses to the comments below.
Please clarify the advantages of using INTEGRA system compared to the other commercial devices.
At this stage, we believe that INTEGRA is an alternative system to those already available on the market. Additionally, the feature is the open structure of the system, the surgeon and the medical center are not forced to use equipment from one manufacturer. This should improve the accessibility of the image-guided systems.
We elaborated that topic on line 322-325.
‘Please consider adding a demographic table.
We added demographic table on line 100.
Please consider adding a paragraph to describe how this intraoperative device worked.
We elaborated that topic on line 135-141.
Please improve the resolution of all figures.
All available figures are 300 dpi, except for intraoperative images. The camera attached to the microscope does not provide such a high resolution. We admit that after building a PDF file before submission the article, some images are blurry. We will bring the Editor on that matter's attention.
Yours faithfully,
The authors
